# Current Status of Cellulosic and Nanocellulosic Materials for Oil Spill Cleanup

**DOI:** 10.3390/polym13162739

**Published:** 2021-08-16

**Authors:** Siegfried Fürtauer, Mostafa Hassan, Ahmed Elsherbiny, Shaimaa A. Gabal, Sherif Mehanny, Hatem Abushammala

**Affiliations:** 1Fraunhofer Institute for Process Engineering and Packaging IVV, 85354 Freising, Germany; 2Mechanical Design & Production Department, Faculty of Engineering, Cairo University, Cairo 12613, Egypt; moustafakhaled14@gmail.com (M.H.); smehanny@googlemail.com (S.M.); 3Mechanical Engineering Department, University of Alberta, Edmonton, AB T6G 2R3, Canada; elsherbi@ualberta.ca; 4Cellulose and Paper Department, National Research Centre, Dokki, Giza 12611, Egypt; shaimagabal@yahoo.com; 5Environmental Health and Safety Program, College of Health Sciences, Abu Dhabi University, Abu Dhabi 51133, United Arab Emirates; hatem.abushammala@adu.ac.ae

**Keywords:** cellulose, nanocellulose, oil spillage removal, aerogels, membranes, fabrics, chemical modification, separation, absorption

## Abstract

Recent developments in the application of lignocellulosic materials for oil spill removal are discussed in this review article. The types of lignocellulosic substrate material and their different chemical and physical modification strategies and basic preparation techniques are presented. The morphological features and the related separation mechanisms of the materials are summarized. The material types were classified into 3D-materials such as hydrophobic and oleophobic sponges and aerogels, or 2D-materials such as membranes, fabrics, films, and meshes. It was found that, particularly for 3D-materials, there is a clear correlation between the material properties, mainly porosity and density, and their absorption performance. Furthermore, it was shown that nanocellulosic precursors are not exclusively suitable to achieve competitive porosity and therefore absorption performance, but also bulk cellulose materials. This finding could lead to developments in cost- and energy-efficient production processes of future lignocellulosic oil spillage removal materials.

## 1. Introduction

### 1.1. Oil Spillage as an Environmental and Economic Challenge

Regularly, more than 35 billion gallons of petroleum oil are transported on ships per annum, and more than 10 million gallons are spilled in marine water, which poses severe jeopardy to aquatic habitats and causes catastrophic consequences for tourism, the economy, and human health [1,2].

In disastrous situations, as in the Gulf of Mexico in 2010, 184 million gallons were released into the ocean [3]. During the Gulf War in Kuwait (1991), 38 million gallons were emptied into Arab Gulf water [4]. Tanker spills like Atlantic Empress in 1978 near the West Indies constituted more than 8.9 million gallons, and the tanker ABT Summer in 1991 near Angola coast spilled more than 8 million gallons [5]. Less severely, natural oil seepages in the Gulf of Mexico exceed 2.1 million gallons, and 0.6 million gallons in California and Alaska offshore regions [6], with a best estimate of 18 million gallons around the globe per annum [7].

Oil inhibits sunlight from underwater photosynthetic activity, which directly disturbs the marine food chain. As a result of the fact that nutrients were set free through the bio- or photochemical degradation of oil, massive amounts of green algae were found, often months or even years after the oil spills. Many creatures like mussels with plankton larvae have difficulty in re-settling as the algae then occupy the region. Coral reefs are susceptible to oil components and often die when oil spills occur near the reefs [5]. Oil droplets act as flypaper for many tiny organisms and entrap those organisms and kill them. Larger creatures may assimilate the oil when they go for their trapped microorganism prey, or they may inhale it and get it stuck on their gill membranes as they swim through the emulsion, which eventually intoxicates the human body (if fed on fish). If they are exposed to high concentrations, they will die; at much lower contamination levels, their physiologies will become impaired [8].

For being cost-effective, petroleum-based materials used for oil spillage clean-up have been widely used during the last few decades. However, their toxic effect in the marine environment—which is more harmful than the oil itself in some cases—has encouraged researchers and engineers to look for eco-friendly biodegradable alternatives which manifest almost zero toxicity and low cost.

### 1.2. Current Solutions and Materials Used for Oil Spill Clean-Up

Several solutions were implemented to solve oil spill (or seepage) problems. Those solutions can be used separately or jointly. Dave et al. [9] categorized solution methods into physical, chemical, biological, and in situ burning. Physical methods comprise containment booms, skimming machines, and adsorbent materials. Chemical methods contain dispersants and solidifiers [9]. The biological method (bioremediation) is simply utilizing microbial activity to biodegrade (assimilate) heavy crude oil components into cell biomass and lighter constituents that are ecofriendly [10]. Thermal solution or in situ burning is also an efficient solution to eliminate oil spills; however, combustion wastes sometimes jeopardize aquatic life [11].

Inorganic natural adsorbents including clay, glass, and sand [12], or synthetic adsorbent including polypropylene, polystyrene, and polyester foams exhibited good performance in oil spill removal [13]. However, lignocellulosic adsorbents still have substantial advantages compared to other materials particularly in terms of friendliness to the marine environment and light weight [14], in addition to other benefits that we will throw light upon in the next section. After conducting comparative analysis, mechanical processing and the application of dispersants followed by bioprocessing (bioremediation) is the most efficient procedure for marine oil spills [9].

### 1.3. Lignocelluloses Are Promising Alternatives

As per sorbent material functionality, lignocellulosic material is a promising candidate to carry out oil sorption tasks. Unlimited resource availability of this versatile biopolymer is roughly calculated to be 1.3 × 10^10^ metric tons per annum [15,16], which directly reflects the low cost of developed lignocellulose solutions. The significantly lower specific weight (0.9–1.2 g cm^−3^) than sand and clay (3.2 and 1.7 g cm^−3^) facilitates the floatability of lignocellulose compared to other materials. Biodegradability is the key advantage in promoting lignocellulose material, composed mainly of cellulose, hemicellulose, and lignin [17], and does not have any toxic effect on marine or terrestrial environments. Cellulose-based materials can be versatile functionalized and modified to alter their properties, including porosity and hydrophobicity (oleophilicity) [18]. The porosity of the materials is a prerequisite to allow their use as adsorbers, to enable incorporation of oil spillages in their matrix. To increase the wettability of lignocellulosic materials, which are hydrophilic by nature, with oil and oil-like liquids, their surface can be chemically modified. Both material properties, porosity and hydrophobicity account for the overall adsorption performance of the adsorption materials. Due to emerging interest of lignocellulosic materials for oil adsorption, several reviews on their production and use have been published recently [2,17,18,19,20,21,22,23,24].

## 2. Introduction to Lignocellulose and Nanocellulose

Lignocelluloses, such as wood and plants, are composites of three polymers: cellulose, hemicelluloses, and lignin [25,26]. Cellulose is a semicrystalline linear homopolymer of glucose units connected via 1,4-β-glycosidic bonds. The degree of polymerization of cellulose can go up to 15 thousand units [19,27]. Hemicelluloses are amorphous branched heteropolymers of several sugars, including glucose, galactose, xylose, mannose, arabinose, glucu- and galacturonic acids. These sugars and sugar acids are connected using a variety of glycosidic bonds [28]. Lignin is an amorphous networked polyphenol of three phenyl propane units that differ on the degree of methoxylation at the meta position [29]. Lignocelluloses’ microstructure is very complex. Initially, around 36 cellulose chains bundle together to form elementary fibrils (diameter of 3–4 nm). These fibrils aggregate together to form sheets of microfibrils with a thickness of 10–30 nm, which are in turn embedded in an amorphous matrix of lignin and hemicelluloses [30]. The three polymers interlock together through lignin-carbohydrate complexes [31]. During pulping, the lignin-carbohydrate complexes are chemically disintegrated, the hemicelluloses and lignin are depolymerized, and cellulose is extracted in the form of 20–40 µm pulp fibers. These fibers are directly used in the paper industry or dissolved and spun for the textile industry [32]. Recently, cellulose fibers have been chemically and mechanically disintegrated to form cellulose nanoparticles (NPs) or nanocellulose (NC) [33].

As shown in Figure 1, there are two primary forms of nanocellulose: cellulose nanofibrils (CNFs) and cellulose nanocrystals (CNCs). CNFs are semicrystalline spaghetti-like NPs with a thickness of 5–50 nm and a length of a few millimeters. They can be produced by the mechanical fibrillation of cellulose fibers using various equipment such as microfluidizers, homogenizers, and micro grinders [34]. A synonymous term for CNF is also MFC (micro fibrillated cellulose). CNCs are highly crystalline rod-like NPs with a thickness of 5–20 nm and a length of a few hundred nanometers [35]. They are produced by the chemical liberation of the cellulose crystallites from the pulp fibers using acid hydrolysis [36,37]. Sulfuric acid is mainly used, but other acids and chemical reagents have also been used [38]. If other mineral acids are used for hydrolysis, and larger particles are obtained, the resulting material is microcrystalline cellulose (MCC). Both forms of nanocellulose have been widely explored in the last two decades due to their attractive properties. A special kind of nanocellulose, which falls under the CNFs umbrella, is bacterial nanocellulose (BNC). Unlike other nanocelluloses, BNC is produced using a bottom-up approach by building up CNFs from simple sugars using bacteria [39].

Nanocelluloses have inherited both the properties of cellulose, such as low density, high mechanical properties, biodegradability, and biocompatibility, and the properties of NPs such as specific surface area and self-assembly [40]. However, nanocelluloses are most famous for the abundance of hydroxyl groups on their surfaces, opening the doors for unlimited possibilities to functionalize them [41]. A wide range of chemical moieties has been placed on the surface of nanocelluloses, including simple molecules and polymers [42]. Due to these exciting properties, nanocelluloses have shown great potential in a wide range of applications [43], including tissue engineering [44], automotive and aerospace industries [45], pharmaceutical formulation [46], water filtration [47], flexible electronics [48], packaging materials [49], lightweight composites and construction materials [50,51] and needless to say oil spillage clean up [52,53]. This has also led to massive interest from academia and industry. More than 3000 articles about nanocellulose were published in 2020 compared to around 200 in 2009. This has also led to the establishment of tens of nanocellulose manufacturing facilities worldwide [42].

## 3. Surface Modification of Cellulose and Nanocellulose

Cellulose and nanocellulose are highly hydrophilic materials due to their chemical nature. Hydroxyl groups exposed to the surface of cellulose fibrils and along the cellulose molecule backbone form intermolecular hydrogen bonds but also interfere with water molecules. The wettability of a surface with any solvent is determined via contact angle measurement of a sessile drop of respective solvent—in the case of water, this is defined as the water contact angle (WCA). The higher the WCA, the more hydrophobic is the wetted surface. The theoretical limits are WCA = 0° (full wettability with water) and WCA = 180° (no wettability with water = super-hydrophobic). In practice, materials with a WCA above 90° are related to hydrophobic materials, such as those with WCA above 150° which are considered super-hydrophobic. For increasing the hydrophobicity of cellulosic and nanocellulosic materials, the surface has to be chemically or physically modified. In the following, some methods are summarized and compared.

### 3.1. Silylation

Silylation or silanization is a ubiquitous, effective, and easy method for surface hydrophobization via grafting. Here the hydroxyl groups of cellulose undergo a condensation reaction with a wide range of silanes. Those silanes are generally alkoxy- (methoxy-, ethoxy-) or chlorosilanes, which have additional organic functionality, representing the hydrophobic part of the molecule (see Figure 2; nomenclature for silanes throughout this review: “hydrophobic chain type” + “number of cleavage groups” + “cleavage group type”-silane, e.g., methyltrichlorosilane, abbreviation MTCS).

The alkoxy- or chloro-functional groups are cleavage groups, forming either the respective free alcohols such as methanol or ethanol, or hydrochloric acid during the condensation reaction. The condensation reaction can be initiated thermally, e.g., via chemical vapor deposition of a chlorosilane onto the surface of the dry aerogel, at temperatures of 70–100 °C. Another possibility for grafting is the hydrolysis of alkoxysilanes with diluted acids and mixing with the cellulose matrix before freeze-drying or impregnating the ready-made aerogel and curing at temperatures up to 120 °C. The hydrophobicity depends on the grafting yield, which is related to the process parameters and the type of organic modification of the silane. For instance, long-chain alkyl residues (C16 [54,55,56,57,58], C18 [59]) show also higher hydrophobicity (WCA = 139–159°) as in short-chain alkyl residues (C8 [60,61], C12 [62]; WCA = 120–150°). However, very high WCAs can be achieved by grafting with methyl-substituted organosilanes (methyltrimethoxysilane, MTMS), probably due to a sterically favorable condensation reaction (WCA = 168.4° [63]). Also, vinyltrimethoxysilane (VTMS) is used [64]. Other silane-based surface hydrophobization agents contain fluorinated alkyls (perfluorooctyltriethoxysilane, PFOTES [65] and perfluorodecyltriethoxysilane, PFDTES [66]); however, the WCAs are similar to those achieved by non-fluorinated silanes (146 and 157°, respectively).

### 3.2. Carbon

Hydrophobic modification in the form of carbon has become increasingly important, since this is considered as a green alternative to silanization. Typically, organic matrices are pyrolyzed under the exclusion of oxygen (e.g., 800 °C under N_2_ [67] or at even lower temperatures, 350 °C [68]), and/or mixed with graphene oxide (GO), which could also be reduced to graphene (e.g., 200 °C under H_2_ [69,70]). In other cases, graphite flakes [71] or carbon nanotubes (CNTs) [72] are directly mixed with the cellulosic matrices before freeze-drying. The hydrophobicity (WCA) of carbonized surfaces is lower than that of silylated ones (WCA = 123–141° [67,72]), but still in a range to have effective wetting with oils and oil-like liquids. The adsorption capacities of such carbon-based adsorption materials can be quite high compared to those solely produced by freeze drying. However, this is more a function of the precursor used than the processing route, since adsorption capacities can vary tremendously [67,69,71,72].

### 3.3. Other Methods

Further strategies to increase hydrophobicity is the incorporation of inorganics (TiO_2_ layers [73] or NPs [74,75]) Cu-NPs [76], Ag-NPs [77], Al_2_O_3_-NPs, surfactants (cetyltrimethylammonium chloride, CTAC [78], SDS [79], stearic acid (SA) [71,80], oleic acid [56,81], epoxidized soybean oil (ESBO) [82]) or other types of grafting molecules (polydopamine PDA and octadecylamine ODA [83,84], poly(*N*,*N*-dimethylamino-2-ethyl methacrylate) PDMAEMA [85], polythiophene [86], acid chlorides [87,88], by carboxymethylation [60] or polymers [89,90]). As with the materials formed by carbonization, there cannot be a general rule deduced to state which hydrophobization method is the best in order to achieve the most beneficial material properties. Many different factors, like precursor type and origin, processing method and processing parameters, and even testing setup, influence the evaluation of the adsorption material performance.

## 4. Processing of Cellulose and Nanocellulose to 3D and 2D Materials

Cellulose and its derivatives, by nature, are among the perfect candidates to constitute aerogel textures. The high content of hydroxyl groups in cellulose chains increases the likelihood of linking with other chains to form a stable 3D gel texture, which is already stable without using a chemical crosslinker [91,92,93]. However, concerning improved mechanical stability, shape recovery, and insolubility, chemical crosslinkers could be beneficial. In the following lines, we will throw light on aerogel fabrication methods.

### 4.1. Sol-Gel Preparation

In a sol, colloidal cellulose particles with sizes ranging from 1 nm to 1000 nm are dispersed in a liquid. A gel consists of a solid, spongy 3D network whose clusters are filled with another liquid [94]. The sol-gel reaction is the process in which material converts from liquid sol phase to solid gel phase (gelation). The sol-gel reaction is the most decisive step in creating a porous 3D web structure in an aerogel. In other words, a sol-gel reaction is a reaction of fixing molecules in the same position via physical or chemical links after moving freely in liquid sol [95,96].

As a result of complex intermolecular and intramolecular hydrogen bond networking in cellulose macromolecules, cellulose is not soluble in water and shows amphiphilic nature, but can be dissolved in various solvents and be regenerated afterwards. LiCl/dimethylacetamide (DMAc) [97], LiCl/DMSO [98], alkali/water/urea [82,99,100,101,102] or thiourea [74,103], ionic liquids (IL), e.g., 1-allyl-3-methylimidazolium chloride (AmimCl) [59,104,105,106] and deep-eutectic solvents (DES), e.g., choline-chloride/urea [107]) are most famous cellulose dissolution systems. Cellulose solvents substantially influence regenerated cellulose properties, regardless of the fabricated material being membrane, hydrogel, or aerogel. After a homogenous solution is maintained, new cellulose chains are regenerated (coagulated) using several coagulants depending on the solvent type; for instance, the NaOH/urea system is regenerated by dilute H_2_SO_4_, the DMAc system is regenerated by ethyl acetate, and some other systems do not need anything but water as a coagulant. A wide variety of membranes, either by casting or spinning, can be fabricated by dissolution/regeneration reaction [108].

### 4.2. Stabilization of the Gel

#### 4.2.1. Chemical Crosslinkers

Chemical crosslinkers covalently crosslink the cellulose backbone of the aerogels to ensure higher mechanical stability, better shape recovery, and less solubility. They are usually mixed with the cellulose sol and activated via heat treatment after the aerogel formation. In many cases, the hydroxyl group of the cellulose is a target for crosslinking (typical examples for chemical crosslinkers see Figure 3). Ethers are formed with homo-biofunctional crosslinkers such as 1,4-butanediol diglycidylether (BDE) [82,109], glutaraldehyde (GA) [106,110,111,112], polymethylsilsesquioxane (PMSQ) [113], or hetero-bifunctional crosslinkers, such as (3-glycidyloxypropyl)trimethoxysilane (GPTMS), which undergoes subsequently self-condensation with its silanol-group [85], or epichlorohydrin (ECH) [57,77,114]. Oxidation of the cellulose pyrane ring with NaIO_4_ results in a highly reactive dialdehyde, which could be crosslinked with chitosan to form the Schiff base [101]. The dialdehyde functionality could also react with hydrazine-functionalized CNCs, originating from carboxylated CNCs via carbodiimide chemistry, forming a hydrazone cross-linkage [115]. Ester linkages can be formed between the hydroxyl group of cellulose and pyromellitic dianhydride (PMDA) [61] or butanetetracarboxylic acid (BTCA) [116,117]. The reactivity of cellulose acetoacetate (CAA) with primary amine groups of 3-aminopropyltriethoxysilane (APTES) [118], or the reaction of the azetidinium groups of polyamideamine-ECH (PAE) and carboxyl groups of CNFs [117,119], was used to form other types of covalent linkage. Another common crosslinker is *N*,*N*′-methylene bisacrylamide (MBA) [100,102], which forms urea-like linkages between hydroxyl groups of different cellulose molecules. Grafted VTMO (vinyl-trimethoxysilane) on nanocellulose surfaces can undergo radical polymerization with azobis(isobutyronitrile) (AIBN) [120].

#### 4.2.2. Polymeric Matrix Stabilizers

For increasing the stiffness and hydrophobicity of formed aerogels, polymers can be used as matrix stabilizers. The main difference between chemical crosslinkers and polymers as matrix stabilizers is that polymers do not necessarily form covalent bonds with the cellulosic matrix, but are mainly physically adsorbed and interact by weaker intermolecular bonds. However, the stabilization of the cellulosic matrices by polymer networks is not less effective than by covalent crosslinkers, since the degree of formation of new covalent bonds with the cellulose is highly dependent on the accessibility of linking groups, which may vary between substrates, depending on their origin, morphology and pretreatment. Some examples for polymer-stabilized cellulosic materials are given as follows: wood fines coated with a poly(methyl methacrylate) (PMMA)-solution in tetrahydrofuran (THF) [121] or polyethyleneimine (PEI) on GPTMS stabilized aerogel [85] are examples of polymer coatings from solutions. Polyvinyl alcohol (PVOH), but also ethylene vinyl alcohol (EVOH) are widely used matrix polymers for cellulosic aerogels since they can be thermally cured and can, due to their chemical similarity to cellulose, be crosslinked with the same crosslinker types [89,111,114,116]. Also, natural polymers such as chitosan are used [70,122]. Matrix polymer coatings can also be formed in-situ, such as polyhemiaminals (PHAs) [90], polydopamine/PEI [58], or polymerization resins of methacrylates (styrene-butyl acrylate SBA, ethylene dimethacrylate EDMA) [123]. Also, metal oxide frameworks (MOFs) [124] or inorganic clay species [125], which are not strictly polymeric but also do not covalently crosslink with cellulose, can stabilize the final aerogel.

### 4.3. Drying Processes for Aerogel Formation

#### 4.3.1. Supercritical CO_2_ Drying

Since CO_2_ has a suitable critical point (304 K, 7.4 MPa) in addition to low cost and high safety, it is a kind of fluid that is commonly used for drying cellulose aerogels [74]. Supercritical drying comprises two successive stages; firstly, supercritical carbon dioxide (scCO_2_) diffuse into the gel pores substituting solvent molecules that will be spilled out of the gel structure; secondly, the gel texture will be dominated by supercritical fluid which exhibit neither surface tension nor capillary action behavior; hence, no structure shrinkage will take place. In non-supercritical drying, shrinkage happens by climbing up liquid through capillary channels, leading to inward forces from the channel walls towards the channel axis, which shrinks the whole texture and induces cracks [126,127].

#### 4.3.2. Vacuum Freeze Drying

Prior to the freeze-drying process, the gel is first cooled below the freezing point of the gel liquid (usually water) afterward; the liquid is eliminated by sublimation (transformation of solid to gas directly, without passing through the liquid phase), which is a major factor in preventing shrinkage texture collapse. Temperature and cooling rate play a drastic role in the crystallization and growth behavior of the newly formed structure, including pore size and distribution [105,128]. For cellulose aerogels, freeze-drying was found to be a more efficient way than scCO_2_. Moreover, rapid cooling to cellulose gels inhibits potential agglomeration of cellulose chains, yielding more coherent structure, which is why treatment with liquid nitrogen is so favored before freeze-drying of cellulose [129]. In some cases, bidirectional freeze-drying is favorable to achieve hierarchical structures with high diffusivity and outstanding mechanical properties [62,130].

### 4.4. Alternative Preparation Methods

In this section a brief overview about alternative preparation methods is given, which cannot be attributed to the previous techniques, but still play a role for preparation of porous lignocellulosic materials. In the easiest case, porous cellulose materials can be found in nature, like cellulose sponges typically used as kitchen wipes, but those have comparable low-specific surfaces [86]. The incorporation of chemical foaming agents (e.g., p-toluenesulfonyl hydrazide) or physical foaming agents (e.g., Na_2_SO_4_) into cellulose solutions were reported for the preparation of aerogels [59]. Electrospinning can be used to produce fibers from a dissolved polymer solution or melt. As cellulose acetate (CA) can be processed into deposited fibers with a very high specific surface [58,89], it could either be used directly as absorber material or regenerated to cellulose by deacetylation with NaOH [89]. Also, hydrothermal treatment is another method to produce heterogenous aerogels in a one-pot-approach, e.g., for the system cellulose/GO/silica/PFDTES [66].

## 5. Performance of Lignocellulosic Materials for Oil Spill Cleanup

### 5.1. Three-Dimensional (3D) Materials: Aerogels and Foams

In this review, 3D materials are defined as materials, which obtain their oil spillage cleanup properties via their volume dimension. They are typically aerogels, foams, or spongy materials and can perform either as absorbers or membranes/filters.

#### 5.1.1. Separation Mechanism

Depending on the surface modification, hydrophilic and hydrophobic 3D materials are reported. Hydrophobic 3D materials are typically absorber -like, which means that the oil is entirely absorbed by the porous material and has to be recovered after cleanup (see Table 1 [2,3,4,5,6,7,8,9,12,13,14,15,16,17,18,20,21,22,23,24,25,27,28,29,30,31,32,33,34,35,36,37,38,39,40,41,42,43,44,45,46,47,48,49,50,51,52,53,54,55,56,57,58,59,60,61,62,63,64,65,66,67,68,69,70,71,72,73,74,75,76,77,78,131,132], and Figure 4A). Alternatively, some hydrophobic 3D materials are also used as filters—here the hydrophobic surface repels the aqueous phase—whereby the oil phase passes through the porous material and is so separated (see Table 2, [54,63,65,75,77,85,133], and Figure 4B). By contrast, hydrophilic 3D materials, which are used as separating membranes, retain the oil phase and let the aqueous phase pass (see Table 3, [58,102,119], and Figure 4C). For those hydrophilic materials, for practical reasons, the oil contact angle is expressed as underwater contact angle (UWCA) of the oil droplet in water instead of the WCA. The comparison of different separation mechanisms is depicted in Figure 4A–C.

#### 5.1.2. Material Performance

##### Comparability of Different Materials

The material performance for 3D materials can be expressed either as (a) maximum absorption capacity regarding oil or non-polar solvents, in the case of static absorption processes, or (b) (flux-dependent) separation efficiencies, expressed as the ratio of removal of oil from water (maximum value 100%). As there is no standardized process to test those parameters, the authors from Table 1 [55,62,72,74,76,83,131,132] used different solvents, different oils, and different flux rates to test their oil spill cleanup materials. Consequently, a direct comparison is difficult. Regarding the absorption capacity, it was found that although the gravimetric capacities (expressed as [g oil/g absorber]) are quite different, the volumetric capacities (defined as [cm^3^ oil per g absorber] = cm^3^ g^−1^) are comparable. This is reasonable since the maximum absorption of any oil is limited mainly by the pore volume of the absorber, so the volume of absorbed oils or solvents should be approximately equal. To define a material-specific, comparable parameter, the average volumetric absorption capacity among five selected oils or nonpolar solvents (*n*-hexane, toluene, chloroform, pump/vacuum oil, gasoline), from which at least one was reported in any experimental work in this review, was calculated. For this, their reported gravimetric absorption capacities were normalized via their respective densities and the means calculated. Although this is an easy approach to generate a comparable material parameter, the absorption capacity is also depending on viscosity and hydrophobicity of absorbed oils and solvents, as well as the absorption conditions such as flux and temperature. To estimate the significance of the average volumetric absorption capacity, also the standard deviation is reported here if more than one solvent could be considered in the calculation (Table 1).

##### Correlation between Adsorption Capacity and Porosity/Density

The average volumetric absorption capacity was compared to material-specific dimensions of the absorber material, such as density (*δ*), porosity (*ϕ*) (see Figure 5A–D), but also BET surface area (S_BET_) and WCA. Volume-related dimensions, such as density and porosity, clearly correlate with the oil absorption capacity. The porosity and the density of a material are related via Equation (1),
(1)$$\phi =\left[1-\frac{\delta_{porous}}{\delta_{non-porous}}\right]\cdot 100\%$$
whereas $\delta_{porous}$ would be the density of the final aerogel, and $\delta_{non-porous}$ is the density of the precursor, e.g., pure lignocellulose before aerogel formation. The higher the porosity of the absorber material, the higher is also the absorption capacity. Below 95% porosity the absorption capacity comes close to zero, which can be considered as the technical limit for those materials (see Figure 5A). The logarithmic absorption capacity is then almost proportional to the porosity, and becomes zero at low porosities (see Figure 5B). Vice versa, for high densities (>50 mg cm^−3^) the absorption capacity approaches the zero value (see Figure 5C,D). These relations can be fitted by an exponential expression (parameters see Table 4).

Regarding the cellulosic raw materials for aerogel preparation one could estimate that the dimensions of precursors (nano- or micrometer sized) would influence the porosity and, respectively, the density of the obtained aerogel. It can be seen from Figure 5A–D that although low density/high porosity aerogels are produced from a nano- and micro sized fibrous precursor (CNF/MFC), this is not exclusively limited to it, since low densities and high porosities are also found in aerogels produced from bulk cellulose and BNC. Vice versa, high density and low porosity aerogels could also be manufactured from CNF/MFC, bulk cellulose and BNC. Aerogels from CNC were located exclusively in the low density/high porosity region, with the lowest reported density of any aerogel in this review, 0.1 mg cm^−3^ [162]. However, CNCs as precursors are rarely reported [114,115,162,163]. Interestingly, cellulose derivatives, such as CA, are also efficient precursors for low density/high porosity aerogels [58,61,89,118]. It has to be assumed that density and porosity of cellulose based aerogels is not mainly depending on the raw materials, but likely on the manufacturing process, e.g., used dispersants or freeze-drying parameters. From economic considerations this could be beneficial, since cheaper precursors such as bulk cellulose could be used, resulting in the same material properties as expensive nano-sized precursors.

##### Correlation between Adsorption Capacity and Surface-Related Properties

Surface-related dimensions, such as S_BET_ and WCA, were also related to the oil absorption capacity. However, although at least a weak correlation was expected, no significant trend could be found for both material parameters. Aerogels with high S_BET_ showed low absorption capacity (e.g., S_BET_ = 405 m^2^ g^−1^, oil absorption capacity = 31.1 cm^3^ g^−1^ [106]) and vice versa (e.g., S_BET_ = 33 m^2^ g^−1^, oil absorption capacity = 415.1 cm^3^ g^−1^ [58]). It is noteworthy that there is a wide variation regarding the S_BET_ even within precursor materials classes, which is most probably attributed to very different synthesis conditions and reactants. The S_BET_ range from 3.4 m^2^ g^−1^ [61]–405 m^2^ g^−1^ [106] for bulk cellulose and derivatives, 3.8 m^2^ g^−1^ [156]–397 m^2^ g^−1^ [81] for nanostructured lignocellulose fibers (CNF/MFC), m^2^ g^−1^ [90]–449 m^2^ g^−1^ [67] for BNC to 23 m^2^ g^−1^ [162]–250 m^2^ g^−1^ [115] for CNC.

A high WCA is a prerequisite to empower cellulosic aerogels as hydrophobic oil absorption materials, therefore most reported aerogels have a WCA > 130°. Similar to the surface-related dimension S_BET_, also no clear correlation was found regarding the aerogels’ WCAs and oil absorption capacities. Although most materials show absorption capacities between 10 and 200 cm^3^ g^−1^ at WCA between 130 and 160°, there are also hydrophobic aerogels reported, which have very high absorption capacities at quite low WCA (WCA = 141°, oil absorption capacity = 369.4 cm^3^ g^−1^ [67]) or low absorption capacities even for the highest found WCA (WCA = 167.5°, oil absorption capacity = 9.5 cm^3^ g^−1^ [78]).

##### Filter-Like 3D Materials

For filter-like 3D materials it was found in general that with increasing flux, also the separation efficiency reduces significantly (see Figure 6 [77]). Fluxes range between 120 [54] to 4200 [85] L m^−2^ h^−1^ for hydrophobic materials and 360 [164] to 22,900 [102] L m^−2^ h^−1^ for oleophobic materials. Separation efficiencies are typically between 98% up to nearly 100%. Particularly some oleophobic materials obtain high flux rates with high separation efficiency (22,900 L m^−2^ h^−1^, 99.8% [102]) and could, therefore, be highly promising as 3D filters.

#### 5.1.3. Recovery and Reusability

Reusability and recovery of 3D materials is vital for enormous applications, as those aspects affect cost and durability. There are three main methods for extracting the absorbed oil after certain use cycles which are mechanical, distillation and extraction. Distillation has a major advantage over other types as it keeps shape consistency after the recovery process. Also, the extraction process is more convenient for preventing hydrophobic layer failure which usually gets damaged by mechanical recovery. The main advantage of the mechanical recovery method is the cheap and easy application. In Table 1, Table 2 and Table 3 the recovery method, the number of reported cycles and the remaining absorption capacity are reported. From the data it cannot be concluded that the recovery method is limited to a kind of material. There is also no information available, if the number of reported cycles represent the last cycle before failure of the absorber, or an arbitrary number related to the experimental setup. Therefore, a comparison of experimental works regarding recovery and reusability is difficult. However, most materials are tested for 10 or more adsorption cycles, with a retaining capacity of 80% and higher. It should be mentioned that the oleophobic 3D materials, which can be used as a filter, show the highest remaining capacity after recovery (95–100%). Some materials also have magnetic properties to collect them more easily from the sea water [54,56,74,81].

### 5.2. Two-Dimensional (2D) Materials: Membranes, Fabrics and Meshes

In the previous section we discussed cellulosic materials which obtained their ability to purify water from oil contamination by their 3D structure. Two-dimensional materials, on the other hand, have a significant extension in two dimensions and therefore have the shape of membranes, fabric or meshes. Membranes are recently used more frequently in different fields, such as seawater desalination, gas separation, water purification, and so on [166]. Membranes with a smaller pore size than emulsions can solve the problem of a large amount of oily waste water treatment in which the simple use of adsorbents not only wastes time but also costs more. The membrane technology is attracting considerable attention with regards to oil removal from wastewater due to its advantages such as high effective oil droplets removal, low energy consumption and required medium temperature.

There are many varieties of fiber having different physical and chemical properties, such as cellulose fibers [167,168], CNFs [169,170,171], electrospun CNFs [172,173]. All of these can be used for membrane construction for specific separation requirements. Membranes of the coated biomass fibers prefabricated by either hydrophilic or oleophilic compounds become impermeable to the counterpart liquids, giving rise to oil–water separation efficiency in a broad spectrum of mixtures. Materials based on cellulose can be not only chemically modified for oil/water separation applications [174,175,176,177,178], but also by different methods such as coatings [167,179,180,181], grafting [182,183], electrospinning [170,172], phase inversion [184], and crosslinking [185,186]. These techniques and materials provide a vital platform to overcome various separation challenges. The different types of substrate, their processing and modification conditions, as well as their properties and separation efficiencies are listed in Table 5, Table 6 and Table 7.

#### 5.2.1. Separation Mechanism

According to their shape, the 2D materials can be used as flow-through separators. For establishing a technical separation process, it is beneficial that the minor phase is retained, and the excess phase passes through the 2D material. Depending on the design of the separation process, the retained phase could either be oil or water, and vice versa the phase which passes through the material (see Figure 7A,B). If the retained phase is the oil phase, those 2D materials are designed as hydrophilic and (super-) oleophobic separators (Figure 7A). The surface tension is, therefore, expressed as UWCA, analogous to hydrophilic/oleophobic filter-like 3D-materials (cf. Table 3). If the retained phase is the water phase (Figure 7B), then the 2D material has to be hydrophobic/oleophilic, the surface tension is then expressed by the WCA, analog to hydrophobic/oleophilic filter-like 3D-materials (cf. Table 2).

#### 5.2.2. Material Performance of Membranes

The membranes derived from environmentally friendly materials, especially from cotton or kapok fibers, show oil/water separation efficiencies of above 99.98%, fluxes ranging from 4000 to 22,200 L m^−2^ h^−1^, and robust performance for regeneration for the filtration of toluene/water mixture and repeated uses. Dual superlyophobic membranes are obtained based on fast and simple methodologies with low cost [167]. Recently, nano-fibrous membranes with special wettability, excellent antifouling property, and reusability have attracted increasing interest for oil/water emulsions separation were modified through coating polydopamine and polyethyleneimine on the membrane surface in an aqueous system by an electrospinning technique [170]. Here, emulsions were separated excellently by PNM@PDA@PEI membrane in the acidic, neutral, and alkaline environments with separation efficiency above 99.1%. In another study, GO@CNF membrane were fabricated in aqueous medium without any toxic regent, suggesting its environmentally friendly nature and cost-efficiency [187]. The membrane obtained exhibited a high separation efficiency, excellent antifouling properties, as well as a high flux for the gravity-driven oil/water separation. 2D layered hierarchical membranes usually have excellent permeability, but can also have anti-fouling performance [181]. If processed using SeP as 1D-modifier and GO and LDH as 2D substrate through layer-by-layer method, they show larger UWCA compared to pure GO and LDH coated membranes. SeP + GO coated membranes have better anti-oil-fouling properties and recycling performance (95.02% and 92.31% for the second and third cycle) and can retain higher water flux after six cycles.

Moreover, the hydrophilicity of nanocellulose-based membranes may be the main obstacle to their application in the separation of oil/water mixtures [188,189,190,191]. All-cellulosic membranes can be used in water cleaning applications, where the capture of oil micro droplets is important due to the affinity that CNCs show to oil under water. Cellulosic environmentally friendly and recyclable composite membranes with CNCs play a significant role in selective oil microdroplets removal from water emulsions with efficiency up to 80% [188]. Low cost and environmentally friendly polysaccharide membranes [177,189] and polysaccharide nanofibers membranes [192,193] can be fabricated with effective separation performances. Thin-film membranes with three kinds of polysaccharide nanofiber barrier layer can have very small diameters (e.g., 5 nm for cellulose nanofibers), which define a small pore size (average diameter about 20 nm) [192]. A CNC/Chitin-nanocrystals (ChiNC)/chitosan (CH) membrane had a typical porous poly-dispersity structure with a primary pore size distribution in the range of 2.9–80 nm. The porous structure, which was created on the surface and within the membrane, enhanced the separation performances [189]. Membranes of CNC/ChiNC/CH can be applied to separate water from water-in-oil emulsions, exhibiting a good separation efficiency and oil flux. In addition, it has an excellent flux recovery property and can be easily cleaned for long-term use. The performance of separation was affected by the thickness of the prepared membrane which directly relates to the concentrations of the complex solutions [189]. A novel dual superlyophobicity membrane with a good reusability, stability under various chemical conditions was developed by surface modification with polypyrrole [194]. The modified membranes had a good separation efficiency (around 99%)) at a low applied pressure difference (−0.9 MPa for emulsion separation and only under gravity for mixtures) for both oil/water mixture and oil/water emulsions. The polypyrrole modified cellulose membranes obtain high flux (over 3000 L∙m^−2^∙h^−1^ for mixtures, over 1000 for oil-in-water emulsions and over 100 for water-in-oil emulsions) [194]. Cigarette filter paper was modified through a dip-coating with dodecanethiol-modified polypyrrole particles for superhydrophobic oil/water separation [180]. The WCA reached 155°, because of the changes in the surface composition and the variation in the surface morphology. Filters with superhydrophobicity could effectively separate various oils and organic solvents with a separation efficiency of 98.8% and a good separation stability. A new route of easy fabrication of a membrane with low-cost and versatile availability of both synthetic and natural starting materials was reported [175]. A superhydrophobic/oleophilic membrane (SHP) was obtained by preparation mechanism, which confirmed different techniques of involving reactions. The pore volume was 0.0066 cm^3^ g^−1^ of final SHP membrane after *n*-dodecanethiol functionalization, particularly for the small pores of the size 18.52 nm. In recent years, several cellulose membranes derived from CA have been widely used in the oil/water separation field due to its abundant sources, environment friendliness, and biodegradability [183,195,196]. CA nanofiber membranes with anti-pollution and self-cleaning abilities can be used as water-removal substances for oil/water mixtures, as well as emulsified oil/water and oil/corrosive aqueous systems, with gravity as the only needed driving force. Membranes possess the highest separation flux 38,000 L m^−2^ h^−1^, and the highest separation efficiency of 99.97% for chloroform/water mixtures [171]. Although many studies used polydopamine (PDA) as a coating on polymeric membranes, PDA particles with an optimal concentration of 0.2 wt% to coat cellulose acetate (CA) to enhance membrane with high porosity, roughness and hydrophilicity have also been used [184]. The membrane delivered the highest pure water flux of ~772 L m^−2^ h^−1^, a high oil rejection rate (93–99%) and improved antifouling properties.

To elucidate a possible correlation between material properties and separation efficiency, we investigated the impact of thickness (Figure 8A) and under-water contact angle (Figure 8B) of the membranes on the volume flux. However, for the 2D materials no obvious relationship could be statistically deduced.

#### 5.2.3. Material Performance of Fabrics

A superhydrophobic surface designed by less-expensive, biodegradable and environmentally friendly route for oil/water separation with fabrics has drawn increasing interest across the scientific community [86,197,198]. However, the applications of hydrophobic surfaces are still hampered by lengthy preparation procedures and high-cost manufacture. Recently, there have been many reports of fabricated non-fluorinated superhydrophobic fabrics with both oil/water separation properties and with antibacterial areas [199], flame-retardant finishing [200], UV shielding [201], self-cleaning [201,202], intelligent self-healing [203] or photocatalytic properties [204] of cellulosic cotton fabric surfaces. a durable superhydrophobic and antibacterial cotton fabric that demonstrated high oil/water separation capacity for a broad variety of oils and organic solvents. The as-prepared multifunctional cotton fabrics have the potential to be used in biomedical bandages or protective clothing that works in unsanitary and moist conditions. Facile construction of multifunctional fabrics with durability still remains a challenge in the past decade.

Modified cotton fabric with ultra-high absorptive capacity have been prepared, which can absorb oil over 12.5 times of its own weight [205]. The cost-effectiveness fabrication strategy and outstanding separation performance of modified fabric, which showing strong water-resistance and high oil/organics affinity, could be applied in harsh circumstances (pH from 2 to 14). Superhydrophobicity has been introduced by surface modification of fibers with mercapto silanes followed by click coupling with methacryl-heptaisobutyl polyhedral oligomeric silsesquioxane (MAPOSS), which could be used as an adsorbent material for removing oil from water [206]. Recently, a robust polyhedral oligomeric silsesquioxane (POSS) based self-cleaning hydrophobic fabric possesses strong stabilities against harsh environments and can deal with various oil-polluted solutions was reported [207]. Therefore, this modified cotton fabric can be of great importance in the preparation of robust hydrophobic filtration materials for oil/water separation. Also cellulose-coated cotton fabric with improved thermal stability, mechanical performance, low cost and high separation performance applications was prepared recently [208]. Coated cotton fabric had capability to separate chloroform-in-water emulsion with a residual rate of 6.8%, which was 10.0 and 6.8 times lesser than those by pristine (68.2%) and pretreated cotton fabrics (46.5%), respectively. The developed fabric had the capability to separate various emulsified oil–water mixtures produced in industry, with increased tensile performance by more than 30%. Its surface area was decreased to 81.85 cm^2^ g^−1^ and the volume of the pores in the 0–10 nm range was always less than 0.020 cm^3^ g^−1^. Both the nitrogen adsorption-desorption isotherm and pore distribution results and SEM observation suggested that the regenerated cellulose filled the pores in the fabric. The formation mechanism of a novel cotton fabric, which was functionalized with super hydrophobicity by immobilization and the in-situ reduction method, in which Cu^2+^ was uniformly immobilized on cotton fabric in [Cu(NH_3_)_4_]^2+^ solution, and then reduced by sodium borohydride (NaBH_4_) and modified by *n*-octadecathiol, was discussed [209]. Cu-fabric owned great chemical stability and reasonable mechanical durability, due to the low wear resistance of cotton fabric, the Cu-fabric surface would be destroyed and turned to hydrophobicity. CuO-coated fabric held superhydrophobicity by further 1-dodecanethiol modification had been fabricated via a simple, scalable, and cost-effective process [204]. The generated CuO nanoleaves were constructed on the surface of the fabric implying excellent photocatalytic activity. About 99.8% methylene blue as model substance was degraded, after being exposed to visible light for 120 min. Modified fabric was used for the separation of several oil/water mixtures and exhibited good recyclability [204]. Fabrication of superhydrophobic cotton fabrics commonly suffers from disadvantages such as extensive usage of toxic and expensive fluorinated adhesives, tedious and time-consuming treatment processes, poor durability and compromised mechanical properties. Therefore, a novel straightforward, cost-efficient, eco-friendly and durable PDMS/calcite composite coating for cotton fabric via one-step was explored, which has great potential in diverse applications including self-cleaning apparel, water-proof wall coverings and continuous oil clean-up [210]. Superhydrophobic cotton non-wovens, which are prepared by atmosphere pressure plasma polymerization in a one-step process, is another approach for efficient separation of oil/water mixtures. A superhydrophobic oil–water separation material was fabricated via an atmospheric pressure plasma jet with cotton non-woven as the substrate [211]. Compared with octamethylcyclotetrasiloxan (D4) plasma-treated cotton, hexamethyldisiloxan (HMDSO) plasma-treated cotton possesses much more stable coating, especially at a lower jet movement speed. Moreover, air/HMDSO plasma-treated cotton non-woven with a jet movement speed of 3 m min^−1^ has excellent self-cleaning ability and could be applied in the oil–water separation with a high separation efficiency and stability towards acid, salt, alkali solution and high temperature [211]. Traditional oil/water separation cotton fabrics were usually fabricated by surface coating with inorganic nanoparticles [212] combined with non-renewable and non-biodegradable or even toxic fossil based chemicals such as fluorine [212], which would lead to secondary environmental pollution after their lifetime. Recently, robust, nanoparticle-free, fluorine-free separation cotton fabric was prepared, which showed excellent mechanical stability and chemical/environmental resistances [197]. Conventional superhydrophobic materials for oil/water separation were usually prepared from non-degradable and non-renewable resources, which would not only increase resource crisis but also cause environmental pollution after being discarded. ZnO/SA-modified cotton fabrics have a biodegradability rate of 59.0% after immersion in phosphate buffer saline solution containing cellulase (pH = 4.8) for 15 days. They also have superhydrophobicity after immersion in acid, alkali solutions or under UV irradiation [202]. However, CNCs with plenty of advantages such as renewability, biodegradability, nanoscale size and low density, were given as an ideal alternative to the inorganic nanoparticles in the creation of rough structure for eco-friendly superhydrophobic oil/water separation materials. CNC has been used to fabricate eco-friendly cotton fabric with great viability as a sustainable and degradable alternative to traditional non-renewable and non-degradable oil/water separation materials [198]. Cellulosic fabric was degradable with weight loss of 14.4 wt% after hydrolytic degradation in phosphate buffer solution (pH 7.4) at 37 °C for 10 weeks. Preparation of CFs with thermal, chemical valve effect and responses to environmental factors by applying a thin layer of copolymer originated from isopropylacrylamide and acrylic acid was reported [213]. Due to the fact that the individual fibers were bonded with each other with the assistance of the polymer coating, this increased the friction and adhesion among fibers, and reduced the defects and weak points in yarn under the mechanical loading. Coated cotton fabrics enhanced their breaking force to 213.5 ± 3.4 N and their density to 140.0 ± 4.2 g m^−2^. Besides the excellent hydrophilicity of coated cotton fabrics, it exhibited selective permeability to water solutions with different pH values and temperatures. The value on the water flux increased by more than 14 times from 0.027 mL cm^−2^ s^−1^ at room temperature to 0.44 mL cm^−2^ s^−1^ at 50 °C, a reflection of excellent response to temperature. More specifically, the highest flux with a value of 0.31 mL cm^−2^ s^−1^ was observed when the solution with pH = 4.0 was poured for filtration, and liquid flux was decreased by more than 90% to 0.021 mL cm^−2^ s^−1^ with increasing pH value of solution till pH = 8.0 [213].

**Table 5 polymers-13-02739-t005:** Cellulose coated mesh for oil removal (values are marked with * if it is the WCA; otherwise the UWCA is reported).

Cellulose Type/Mesh Material	Hydrophobic Modification	Pore Diameter [µm]	Porosity [%]	WCA */UWCA [°]	Flux [L m^−2^ h^−1^]	Separation Efficiency [%]	Number of Recovery Cycles	Ref.
hydrogel								
--	--	50	--	151	12,885	98.2	60	[214]
nylon	hydroxylation			150	15,840	99.99	12	[215]
stainless steel	--	80	72	156	38,064	98.9	10	[216]
nanocrystals								
stainless steel	silanization	175	75	163 *	--	95	40	[217]
copper	hydroxylation	25	--	155	35,000	--	12	[218]
filter paper	hydroxylation	20	--	150	4000	--	12	[218]
nanofiber								
steel	silanization	150	61	160 *	--	99	50	[219]
filter paper	silanization	220	53	156 *	38,064	98.9	--	[220]
acetate								
cellulose	--	280	--	--	400,000	96	--	[221]
cellulose	hydroxylation	280	73	--	160,000	99	20	[222]
polyamide	plasma treatment	64	--	155	--	99.6	25	[223]
composite fiber	cellulose/polymer composite	20	84	--	1910	97.3	10	[224]

**Table 6 polymers-13-02739-t006:** Cellulose based membranes for oil removal (values are marked with * if it is the WCA; otherwise the UWCA is reported).

Membrane Material/Cellulose Type	Hydrophobic Modification	Pore Diameter [µm]	Membrane Thickness [µm]	Porosity [%]	WCA */UWCA [°]	Flux [L m^−2^ h^−1^]	Separation Efficiency [%]	Number of Recovery Cycles	Ref.
Cellulose									
CA	polymerization	360	--	--	155 *	3000	99	3	[194]
CA	--	470	303	45.1	154 *	20	--	--	[172]
CA	--	--	18.8	67	162 *	3106	99	10	[225]
CA	--	40	50	--	161 *	--	--	--	[195]
eucalyptus pulp	grafting	--	--	--	130 *	--	97.6	5	[182]
filter paper	chemical modification	18	--	--	152 *	960	96.7	20	[175]
filter paper	hydroxylation	--	--	--	150 *	70	98.7	10	[186]
filter paper	grafting	--	1000	--	160 *	2350	--	5	[176]
nanofibers	spray coating	--	--	--	144 *	--	94.5	7	[169]
CA	screen printing	--	450	--	150	--	83	--	[188]
CA	deacetylated	--	--	--	137	38,000	99.9	50	[171]
CA	--	12	83	79	--	772	99	--	[184]
CA	wet phase-inversion method	--	--	--	157	20	99.4	--	[177]
CA	ultrasonication	70	250	--	--	1217	93.2	--	[226]
CA	--	100	--	--	--	435	99.8	3	[196]
CA	--	150	--	70	--	97,200	98	--	[227]
CA	--	450	--	--	151	1910	98	6	[181]
CA	grafting	--	240	--	--	110	100	3	[183]
filter paper	chemical modification	--	--	--	140	750	--	4	[174]
nanocrystals	vacuum-assisted filtration	70	0.6	--	176	1734	90	10	[190]
nanofibers	--	20	0.1	--	--	272	99.5	3	[192]
nanofibers	grafting	--	--	--	166	200	99.1	20	[170]
nanofibers	deposition	129	--	--	165	3730	99	--	[228]
nanofibers	--	--	100	83	155	960	97.3	10	[187]
nanofibers	grafting	--	--	--	160	--	--	5	[229]
powder	--	--	0.112	--	150	1620	92.5	--	[230]
--	--	50	54	83.2	--	4000	90	--	[193]
--	--	160	--	86	--	26	99.9	--	[231]
cotton									
fibers	polydopamine (PDA) nanoparticles	--	--	--	150	22,200	99.98	10	[167]
linter pulp	lower critical solution temperature (LCST) system	312	--	--	165	200	99	10	[168]
fabric									
nanocrystals	--	--	220	--	134	--	98	3	[191]
nanocrystals	vacuum-assisted filtration method	80	95	43	153	99	99.2	6	[189]
filters									
cigarette filters	dip-coating	--	--	--	155 *	--	98.8	30	[180]
methylcellulose	chemical bath deposition method	--	90	--	150 *	1055	--	10	[178]
qualitative filter paper	combining in growth technique	15,000	340	--	155 *	537	94.4	50	[232]
qualitative filter paper	SA	15,000	340	--	154 *	--	95.1	50	[233]
EC	ultraviolet induced crosslinking	--	--	--	--	4332	--	--	[185]
paper fiber	deposition on the surface by alternating soaking process (ASP)	--	0.22	--	162	550	99	20	[179]
peanut shell	--	--	51	--	--	--	--	--	[234]

**Table 7 polymers-13-02739-t007:** Cellulose fabrics for oil removal (values are marked with * if it is the WCA; otherwise the UWCA is reported).

Cellulose Type/Fabrics Type	Hydrophobic Modification	WCA * [°]	Flux [L m^−2^ h^−1^]	Separation Efficiency [%]	Recovery Method **/Cycles/Capacity after	Ref.
Cotton						
plain weave fabric	silanization	151 *	--	98	DW/10/95%	[199]
raw fabric	silanization	164 *	--	99.95	E/40/99.5%	[205]
raw fabric	silanization	142 *	74,161	96	DW/50/96%	[207]
raw fabric	silanization	159 *	114,744	99.5	UW/--/--	[206]
raw fabric	silanization	152.7 *	30,000	99	--/10/99.4%	[235]
raw fabric	salt solution	150 *	4000	93.2	--/--/--	[208]
raw fabric	silanization	138 *	--	99	W/7/95%	[200]
raw fabric	Cu nanostructured	150 *	--	98	FW/30/97%	[209]
raw fabric	Cu nanostructured	151.5 *	--	96	E/20/--	[204]
raw fabric	silanization	150 *	--	99	SC/40/99%	[210]
nonwoven fabric	silanization	155 *	--	97	SC/100/97.5%	[211]
microfibers	silanization	150 *	--	94	E/7/94%	[236]
raw fabric	silanization	150 *	1749.7	97.9	SC/100/97.5%	[197]
raw fabric	silanization	160 *	--		SC/20/--	[201]
raw fabric	grafting	164 *	--	96.5	SC/10/--	[202]
raw fabric	silanization	154 *	2688	96.2	E + DW/10/92%	[237]
raw fabric	isopropylacrylamide and acrylic acid	160 *	--	99.5	--/4/99.5%	[213]
raw fabric	silanization	151.5 *	--	95	E/20/98%	[238]
cellulosic						
raw fabric	silanization	150 *	41,800	90	DW/4/89%	[239]
raw fabric	silanization	164 *	53,000	98	--/10/98%	[198]
raw fabric	silanization	155 *	3400	98	DW/20/--	[240]
pristine						
textile fabric	lauric acid (LA)-TiO_2_ composites and Fe_3_O_4_-NPs	153 *	5700	99	E + DW/25/98%	[241]
textile fabric	silanization	154 *	4500	97	E/50/97%	[242]

Sorted with decreasing separation efficiency; *, in brackets: source or derivative; **, recovery methods: deionized water (DW), ethanol (E), ultrasonic washing (UW), water (W), flowing water (FW), self-cleaning (SC).

## 6. Conclusions and Recommendations

Extensive efforts have been made to investigate the potential of lignocellulosic materials for oil spill removal. In particular, a multitude of cellulose- and nanocellulose-based materials have been explored, which underwent a wide range of chemical modifications to enhance their hydrophobicity, including silanization with a variety of different silanes, grafting of polymers and hydrophobic molecules, and incorporation of inorganic nanoparticles and surface modifiers. Physical modification for hydrophobization has been used as well including carbonization via pyrolysis. The used materials were mostly 3D such as hydrophobic and oleophobic sponges and aerogels or 2D such as membranes, fabrics, films, and meshes. Regarding the 3D materials, there was a clear correlation between the material properties (mainly porosity) and their performance. In more detail, absorption increases exponentially with lower density (higher porosity). The trend can be observed over all dimensions of cellulosic precursors (cellulose, nanocellulose, cellulose derivatives). It is worth mentioning that nanocellulosic aerogels did not always yield higher porosity (lower density) than other cellulosic aerogels. The preparation conditions and techniques used together were the determinants of the materials properties; hence their performance. This discourages the use of nanocellulose due to its high cost since its nanosize did not add a value to the material performance (absorption capacity). Regarding the 2D materials, no clear correlations between the material properties (thickness and surface properties) and its performance were observable. Trends in production are the introduction of hierarchical porosity for fast adsorption and desorption processes while maintaining high surface area, increased mechanical stability for efficient recovery of adsorbed oil, and several life cycles of the absorber materials, and development of cost-efficient and reproducible processes while using less-toxic and environmental-friendly modifiers. Overall, cellulosic materials have shown promising capabilities to clean up oil spillage, although more thorough investigations should be performed regarding the impact of the material preparation conditions and techniques on its properties and performance, especially for 2D materials.

## Figures and Tables

**Figure 1 polymers-13-02739-f001:**
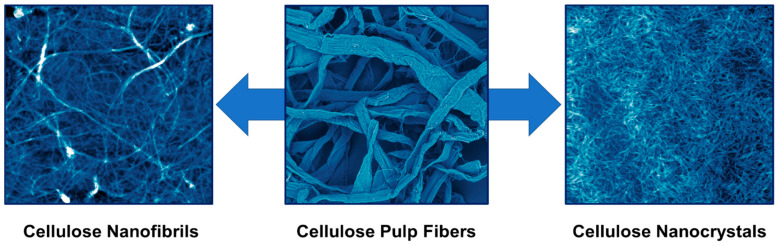
From cellulose pulp fibers to CNFs and CNCs. The photo of the cellulose pulp fibers was obtained using a scanning electron microscope (500 × 500 µm), while those of CNFs and CNCs are atomic force microscopy images (5 × 5 µm).

**Figure 2 polymers-13-02739-f002:**
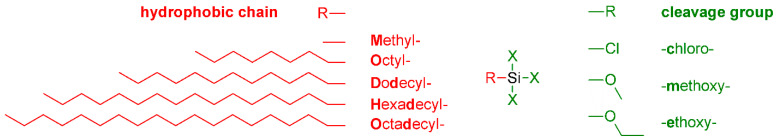
Typical functional groups for hydrophobic silanes.

**Figure 3 polymers-13-02739-f003:**
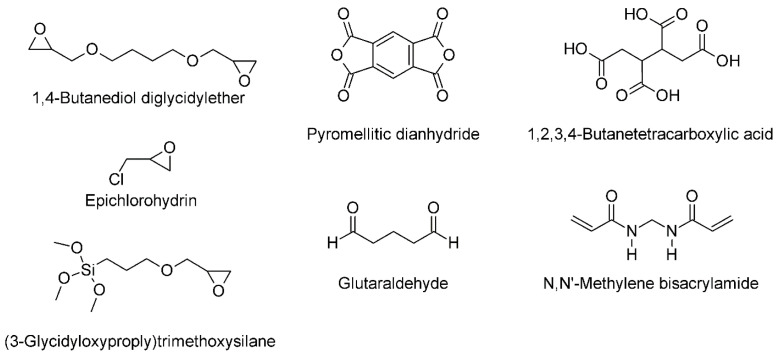
Typical molecules for cellulose crosslinking.

**Figure 4 polymers-13-02739-f004:**
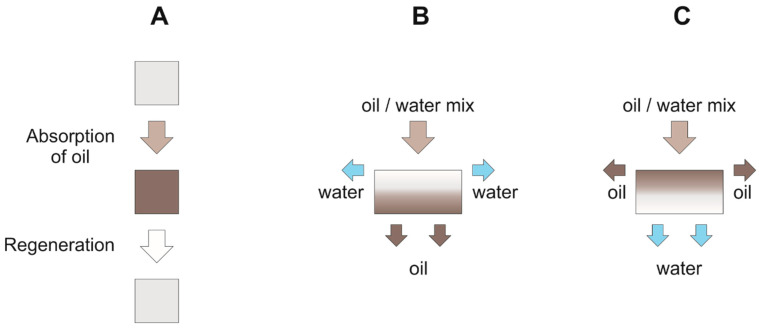
Separation mechanism of 3D materials: (**A**) absorber type; (**B**) hydrophobic filters; (**C**) hydrophilic filters.

**Figure 5 polymers-13-02739-f005:**
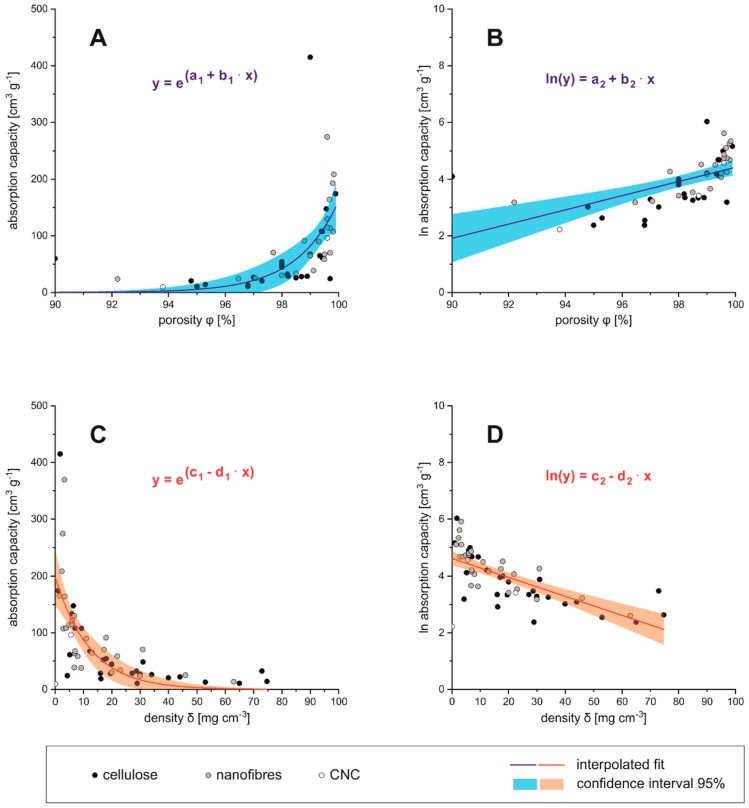
Density (**A**,**B**) and porosity (**C**,**D**) in relation to absorbance capacity.

**Figure 6 polymers-13-02739-f006:**
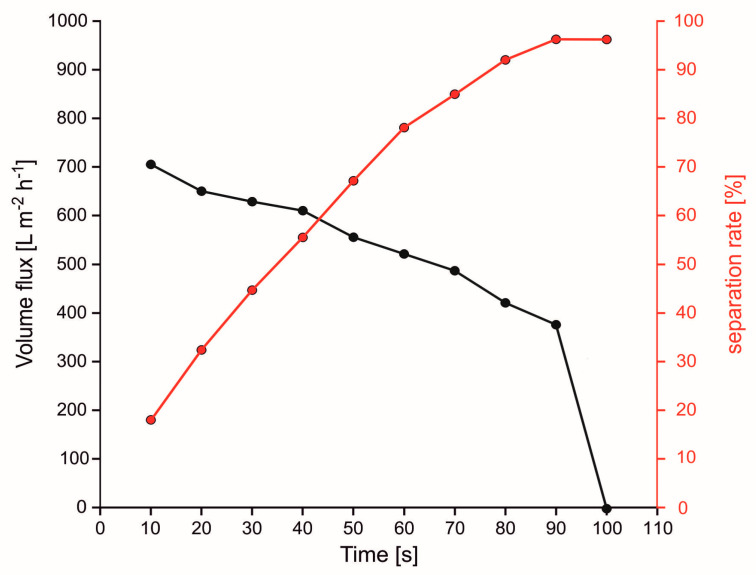
Decreasing separation efficiency (rate) with increasing flux (adapted from [77]).

**Figure 7 polymers-13-02739-f007:**
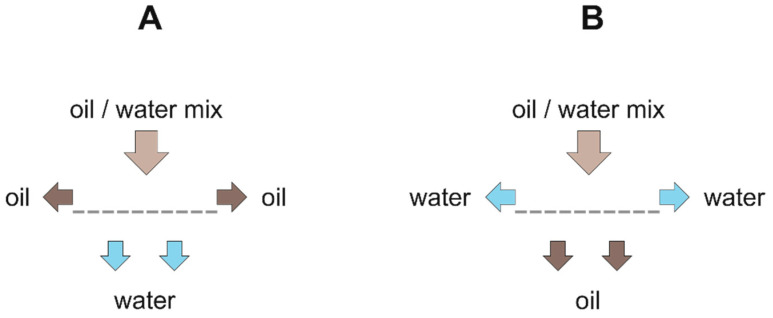
Separation mechanism of 2D materials: (**A**) hydrophilic/oleophobic filters; (**B**) hydrophobic/oleophilic filters.

**Figure 8 polymers-13-02739-f008:**
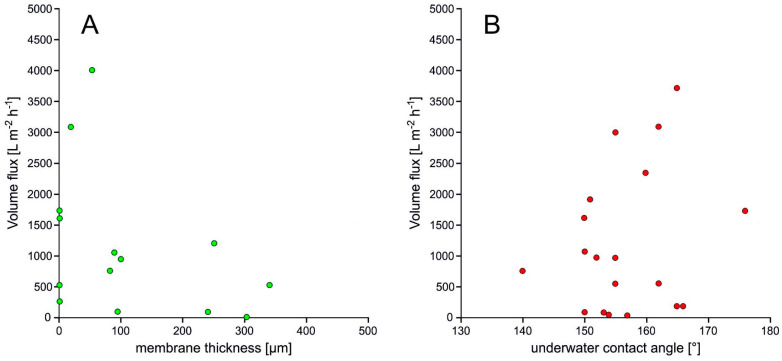
Volume flux of membranes depending from membrane thickness (**A**) and UWCA (**B**).

**Table 1 polymers-13-02739-t001:** Hydrophobic aerogels used for oil absorption.

Matrix Material *	Hydrophobic Modification	S_BET_ [m^2^ g^−1^]	δ [mg cm^−3^]	WCA [°]	Oil Removal Rate [cm^3^ g^−1^]	Recovery Method **/Cycles/Capacity after	Ref.
Cellulose							
(CA) + graphene	HDTMS	33	18.4	159.0	415.1 ± 38.8	M/10/95%	[58]
C. (electrospun)	MTCS	7.9	1		174.1 ± 9.5	M/10/80%	[134]
(CA) + graphene	PVOH		6.45	141.2	147.2 ± 10.9	M/10/80%	[89]
GO	DDTES	47.3	5.9	150.0	133.2 ± 17.2	M,D/10/90%, 100%	[62]
(bamboo) + MnFe_2_O_4_-NP	NR	182	NR	153.0	119.8 ± 5.3	M/10/70.5%	[135]
(paper waste)	Kymene/MTMS	NR	7	153.5	108.0	NR	[136]
(sisal) + Cu-NP	NR	NR	4.56–11.23	150.3	107.4 ± 0.4	D/10/85%	[76]
(CA butyrate) + EVOH + GA	NR	NR	12.3	145.0	67.1 ± 6.8	D,M/10, 15/100%	[110]
(wood fines) + PMMA	Alkyl-ketene dimer	NR	13–127	135.0	64.8	NR	[121]
(cotton, paper waste)	MTMS	NR	5.13–8.5	142.8	61.1	D, M/5	[137]
(CAA) + CNF + APTES	alkylamines	NR		146.0	59.8 ± 4.0	-/10/85, 90%	[118]
(ethylcellulose EC) + oleic-acid coated Fe_3_O_4_ –NP	HDTMS	NR	<18	152.8	54.3 ± 3.1	D/50/88%	[56]
(Corn stalk pits)	ODA	2.7	NR	157.2	49.8 ± 6.1	E/20/80%	[84]
(cotton) + (MBA)	MTCS	NR	31	141.0	48.2 ± 12.9	M/10/70%	[100]
(EC) + ECH/Si-CNT/nanosilica	HDTMS	NR	<20	158.2	44.5 ± 1.9	D/50/86%	[57]
(paper waste)/GA	TMCS	405	28.7	135.0	32.1 ± 2.5	NR	[106]
(hardwood)/BDE	ESBO	NR	73	132.6	32.1 ± 0.2	M/30/90%	[82]
(reg. MCC) + Fe_2_O_3_-NP	TiO_2_ (sol-gel)	NR	NR	NR	31.8	E	[74]
(cotton)	MTMS	NR	NR	154.0	29.2 ± 1.0	18/90.6%	[138]
(waste paper)	MTMS	NR	27.2	146.1	28.4	M/10/60%	[139]
(sugarcane bagasse)	MTMS	NR	16–112	150.5	28.4 ± 2.5	NR	[140]
(cotton)	MTCS	NR	19.6	150.0	28.1 ± 1.2	D/15/100%	[99]
(balsa wood)	MTMS	23.4	30	151.0	26.7 ± 1.4	M/10/94%	[141]
(bamboo, cotton)	ODTMS	NR	34	156.0	25.8 ± 3.0	E/30/	[59]
(C. diacetate) + PMDA	OTCS	3.4	4.3	120.0	24.2 ± 7.1	M/10/40%	[61]
(wheat straw)	TMCS	36–143	44–150	136.0	21.8	NR	[142]
NR	MTMS	NR	40	145.0	20.5	M/5	[131]
(paper waste)	MTMS	NR	40	135.3	20.5	M	[132]
+butanediol diglycidyl ether crosslinked (BDE)	NR	NR	14–24	NR	18.5	M/30/18%	[109]
(balsa wood)	PDMS	NR	74.8	150.0	13.8 ± 0.5	M/18	[143]
+chitosan	TMCS	NR	53	152.8	12.7 ± 9.0	M/50/83%	[101]
(foam) + PE, PP, graphite	SA/graphite	NR	NR	145.0	11.3 ± 1.8	M/15/40–60%	[71]
(paper waste)	MTCS	NR	29	136.0	10.7	M/5	[105]
+chitosan	sodium stearate	NR	65	156.0	10.7	E/5	[80]
(cotton)	DMAEMA	NR	NR	130.0	9.8 ± 2.4	-/5	[144]
(MCC)	CTAC	NR	NR	167.5	9.5 ± 0.9	M/20	[78]
(natural sponge)	polythiophene	NR	NR	126.6	6.8 ± 0.3	-/5/70%	[86]
CNF/MFC							
(Rice Straw)	OTES	10.9	1.7–1.8	NR	274.6 ± 18.8	D/6/48%	[145]
MOF	MTMS	NR	9	150	209 ± 49	M/25/86%	[124]
NR	MTMS	NR	2.4	154.0	208.2 ± 14.6	M/30/100%	[146]
(bamboo leaf)	MTMS	NR	NR	152.0	205.1	NR	[147]
(cotton) + nanochitosan	reduced GO	110	9.3	115.3	186.1 ± 17.2	M/10/92%	[70]
(cotton)	SDS	151	1.5	NR	165.0	NR	[79]
(soft wood Kraft pulp)	MTES	94.8–195.5	3.41–5.08	151.8	164.0 ± 29.3	E/30/65%	[148]
(kapok)	VTMS	NR	5.1	140.1	149	E/10/87%	[64]
(rice straw) + (PAE)	MTMS	178.8	2.2	151	135.7 ± 19.7	M/5/45%	[117]
BTCA	HTMS	NR	6	151	134 ± 15	E/30/78%	[149]
clay sepiolite	MTMS	NR	6	128	126.8 ± 33.5	M	[125]
NR	PDA/ODA	93.1	6.04	152.5	121.6 ± 3.7	M/1/50%	[83]
+PVOH, BTCA	MTCS	35.1–106.1	4.66–16.54		114.5 ± 12.1	NR	[116]
(reed)	MTCS	55.2	4.9	155.0	113.6	M/5/80%	[150]
(recycled waste fibers)	MTMS/HDTMS	NR	2.9	150.0	106.9 ± 7.3	M/30/71–81%	[107]
(bamboo)	MTCS	NR	17.95	142.0	90.9	NR	[151]
	HDTMS	>261.9	11–17.5	138.9	89.8	M/20	[55]
CNF + Al_2_O_3_-NP	NR	124	5.1	NR	84.3 ± 15.5	NR	[152]
(oat straw pulp powder)	MTMS	25	17.3	136.0	69.9 ± 2.8	E/10	[153]
(mango wood)	Stearoyl chloride	156	7–20	159.0	67.5 ± 2.2	M/15/75%	[87]
(eucalyptus) + PVOH, GA	MTCS	172	13	150.0	64.4 ± 3.3	NR	[111]
	MDI	228	6.9, 8.3	NR	63.6	E/5	[154]
(carboxymethylated)	OTCS	11–42	4–14	150.0	58.4	NR	[60]
+GO, silica	PFDTES	29.5–93.5	10–40	155.5	58.4 ± 8.1	D/10/100%	[66]
	stearoyl chloride	NR	NR	160.0	45.2 ± 3.9	M/10/56%	[88]
(bamboo)	TMCS		6.78	117.0	38.6	NR	[155]
+Fe_3_O_4_	oleic acid	397	9.2	84.5	37.8	NR	[81]
(recycled waste fibers) + SBA, EDMA	Kymene	18.4	23	149.0	34.1 ± 2.1	D/5/40%	[123]
	TiO_2_ (ALD)		20–30	90.0	30.2 ± 2.1	D/10/100%	[73]
(furniture waste)	MTMS	3.8	46	138.8	25.1	NR	[156]
	MTCS	NR	NR	148.7	24.0 ± 2.1	M/10/92.4%	[130]
(eucalyptus, pinus)	MTMS	NR	30	134.0	23.9	NR	[157]
(pulp waste)	MTMS	NR	NR	133.5	19.1	NR	[158]
(eukalyptus)	carbonization	NR	112	130	7.4 ± 0.8	D/100/95%	[68]
(pine needles)	MTCS	20.1	3.12	135.0		NR	[159]
BNC							
	carbon	449	3.3	141.0	369.4 ± 57.7	M/10/94.6%	[67]
+GO	graphene	NR	NR	NR	192.6	NR	[69]
	MTMS/PMSQ	NR	0.7	168	148.8 ± 13.0	M/10/90%	[113]
	TMCS	180.7	6.77	146.5	129.7 ± 7.2	M, E/10	[160]
	CNT	NR	3.9–10.8	123.3	108.0	NR	[72]
	PHA	46.5	30.9	143.0	70.5	E/3	[90]
+silica	MTES	NR	63	152.0	13.5 ± 1.9	M	[161]
CNC							
+hydrazine	NR	250	5.6		96.0	M/20/80%	[115]
+PVOH, ECH	MTCS	38	22.5–36.1	114.9	30.4 ± 7.1	M, E/10/90%	[114]
+polysilsesquioxane	IPTES	23–90	0.11–0.17		9.2	NR	[162]
	CPTES		86		NR	E/10	[163]

Sorted with decreasing oil removal rate, *, in brackets: source or derivative; **, recovery methods: Extraction (E), distillation (D), mechanical (M); Not Reported (NR).

**Table 2 polymers-13-02739-t002:** Hydrophobic aerogels used as filters.

Matrix Material *	Hydrophobic Modification	S_BET_ [m^2^ g^−1^]	δ [mg cm^−3^]	WCA [°]	Flux [L m^−2^ h^−1^]	Separation Efficiency [%]	Recovery Method **/Cycles/Capacity after	Ref.
Cellulose								
+SiO_2_-NP, TiO_2_-NP	APTMS	230	135	138.5	667	99.99	NR	[133]
(EC) + ECH + Ag-NP	*N*-dodecyl mercaptan		17	161.3	NR	99.76	D/50/90%	[77]
	TiO_2_ (sol-gel)/OTMS		NR	171.0	NR	98.50	E/40/93%	[75]
+Fe_3_O_4_-NP	HDTMS		NR	156.0	120	98.00	E/5	[54]
(filter paper)	PFOTES			146.0			M/30	[65]
CNF								
+GPTMS + PEI	PDMAEMA	12.72	57	130.0	4200	99.96	NR	[85]
CNF + SiO_2_-NP	MTMS	108.6	<6.43	168.4	2000	99.50	E/20/100%	[63]
BNC + GA + Ag-NP	silane-based zwitterionic compound		13	153.0	NR	99.22	-/10	[112]

Sorted with decreasing oil removal rate, *, in brackets: source or derivative; **, recovery methods: Extraction (E), distillation (D), mechanical (M); Not Reported (NR).

**Table 3 polymers-13-02739-t003:** Oleophobic aerogels used as filters.

Matrix Material *	Hydrophobic Modification	S_BET_ [m^2^ g^−1^]	δ [mg cm^−3^]	UWCA [°]	Flux [L m^−2^ h^−1^]	Separation Efficiency [%]	Recovery Method **/Cycles/Capacity after	Ref.
Cellulose								
(cotton) + MBA + GO	GO		10	NR	22,900	99.80	M/10/99%	[102]
(CA) + graphene	polydopamine/PEI	33	18.4	143.5	NR	NR	M/10/95%	[58]
CNF/MFC								
	mercaptopropionic acid		NR	147	NR	100.00	E/10/95–100%	[120]
	sulfonation		52.24	160	360	99.97	E/20/100%	[164]
+chitosan	NR		18.6	160	NR	99.00	-/40/98.6%	[122]
+PAE	NR		NR	155.6	2405	98.60	-/10/100%	[119]
+chitosan	NR		6.1	143	NR	NR	-/16	[165]

Sorted with decreasing oil removal rate, *, in brackets: source or derivative; **, recovery methods: Extraction (E), distillation (D), mechanical (M); Not Reported (NR).

**Table 4 polymers-13-02739-t004:** Fit parameters to describe the relation between porosity or density and absorption capacity.

Related Figure	Constant Term	Factor
Figure 5A	a_1_ = −62.29 ± 15.37	b_1_ = 0.67 ± 0.15
Figure 5B	a_2_ = −20.62 ± 4.98	b_2_ = 0.25 ± 0.05
Figure 5C	c_1_ = 5.29 ± 0.12	d_1_ = 0.08 ± 0.02
Figure 5D	c_2_ = 4.62 ± 0.13	d_2_ = 0.03 ± 0.005

## Data Availability

Not applicable.

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
