# Peer review of "Current Status of Cellulosic and Nanocellulosic Materials for Oil Spill Cleanup"

_polymers, 2021, doi:10.3390/polym13162739_

Round 1

Reviewer 1 Report

Dear Authors manuscript entitled Current Status of Cellulosic and Nanocellulosic Materials for Oil Spill Cleanup is well written summary of actual knowledge about methods for valorization of cellulosic and nanocellulosic materials for application in process of cleanup common environmental pollutant as oil spill. The review is suitable to be published in Polymers Journal.

Comment;

The bracket with citations should be placed on the end the sentence. When authors use a few citations, more than two, they should be written as multi citations e.g  [43-47].

Reviewer 2 Report

The paper is well prepared, written, and formatted. There are no significant shortcomings in it. The only recommendation relates to the use of the up-to-date literature for discussion. In the present state, there are several references from 2020, while no literature from 2021. It is suggested to enhance the present manuscript with 2020-2021 literature discussion. 

Reviewer 3 Report

Siegfried et al. present a review of the current research status of cellulosic and nanocellulosic materials for oil spill cleanup. The surface modifications, basic preparation techniques, the morphological characteristics and the related separation mechanisms of lignocellulosic substrate materials, and their properties have been carefully summarized. This review provides a certain reference for the development to cost- and energy-efficient production processes of future lignocellulosic oil spillage removal materials. I think this paper can be accepted after fully addressing several major issues as noted below:

Introduction:

  1. Page 1, line 35: “[1][2]” should be “[1,2]”. There are many similar errors, such as page 2, line 81; page 4, line 169-170, etc., please check the full text.
  2. Page 2 “2. Current Solutions and Materials Used for Oil Spill Clean-up”, the content of this part is too short, please briefly summarize the advantages and disadvantages of each method or summarize it in a list.
  3. The second paragraph of “2. Current Solutions and Materials Used for Oil Spill Clean-up” on page 2 lacks a transition from the previous paragraph, and the advantages of the natural adsorbents illustrated are not highlighted compared to the previous paragraph.
  4. Page 2, line 78: what does “high-performing” mean? Please elaborate further.
  5. Page 2, line 87: " porosity and hydrophobicity", please further elaborate on the advantages of porosity and hydrophobicity.

Introduction to Lignocellulose and Nanocellulose

  1. Page 2 “ Introduction to Lignocellulose and Nanocellulose”, does not highlight the advantages of cellulose in oil spill cleanup. Here, the structure of cellulose should be briefly introduced, and highlight the potential of cellulose as oil spill cleanup based on its structural advantages and characteristics. Page 3, line 124-137, the introduction about nanocellulose has the same problem.
  2. Page 3, line 120: what does “attractive properties” mean? Please elaborate further.

Surface Modification of Cellulose and Nanocellulose

  1. In “2 Carbon” and “3.3 Other Methods”, the described methods are not compared, nor the modification mechanism of carbon and other substances is described in depth, so the content is relatively thin.

Processing of Cellulose and Nanocellulose to 3D and 2D Materials

  1. Page 5, “2. Stabilization of the Gel”: How do chemical crosslinkers and polymer matrix stabilizers affect the stability of gels in the same way or differently? What are their advantages and disadvantages?
  2. Page 7: “4. Alternative Preparation Methods” is too short. You can briefly describe the usage scenarios and the advantages and disadvantages of the listed methods. In addition, the title of this section is “Processing of Cellulose and Nanocellulose to 3D or 2D Materials”, but why spend a lot of space on the preparation of aerogels by the sol-gel method, and the preparation methods of 2D materials?

Performance of Lignocellulosic Materials for Oil Spill Cleanup

  1. Page 8, “1.2. Material Performance”, in this part, I would suggest to add subheadings according to different performances to elaborate to make the review more logical. Page 15, “5.2.2. Material Performance of Membranes” and page 17, “5.2.3. Material Performance of Fabrics” have the same problem.

Conclusions and Recommendations

  1. This part should focus on summarizing the challenges of cellulose and nanocellulose in oil spill cleanup applications and the directions for further research in the future, rather than a simple summary of the previous article.

References

  1. The format of the references is wrong, please correct. e.g., “[1] M. Radetic, V. Ilic, D. Radojevic, R. Miladinovic, D. Jocic, and P. Jovancic, “Efficiency of recycled wool-based nonwoven material for the removal of oils from water,” Chemosphere, vol. 70, no. 3, pp. 525–530, 2008.” should be “[1] Radetic, M.; Ilic, V.; Radojevic, D.; Miladinovic, R.; Jocic, D.; Jovancic, P. Efficiency of recycled wool-based nonwoven material for the removal of oils from water. Chemosphere, 2008, 70, 525–530.”
  2. The authors could add the following references which would again increase the interest to general functional cellulosic material readers: Journal of Bioresources and Bioproducts, 2020, 5(4): 223-237; ACS Applied Materials & Interfaces, 2021, 13, 7617-7624; Journal of Bioresources and Bioproducts, 5(2): 79–95.
